# Visualization of Accessory Lymphatic Pathways, before and after Manual Drainage, in Secondary Upper Limb Lymphedema Using Indocyanine Green Lymphography

**DOI:** 10.3390/jcm8111917

**Published:** 2019-11-08

**Authors:** María Elena Medina-Rodríguez, María de-la-Casa-Almeida, Efrén Martel-Almeida, Arminda Ojeda-Cárdenes, Esther M. Medrano-Sánchez

**Affiliations:** 1General Hospital of Gran Canaria Dr. Negrin, Barranco de la Ballena, s/n. 35010 Las Palmas de Gran Canaria, Spain; 2Department of Medical and Surgical Sciences, University of Las Palmas de Gran Canaria Las Palmas, Calle Juan de Quesada, s/n, 35001 Las Palmas de Gran Canaria, Las Palmas, Spain; mariaelena.medina@ulpgc.es; 3Department of Physiotherapy, University of Seville, C/Avicena s/n. 41009 Seville, Spain; 4General Hospital of Gran Canaria Dr. Negrin, Barranco de la Ballena, s/n. 35010 Las Palmas de Gran Canaria, Las Palmas, Spain; emaralm75@gmail.com; 5General Hospital of Gran Canaria Dr. Negrin, Barranco de la Ballena, s/n. 35010 Las Palmas de Gran Canaria, Las Palmas, Spain; armindaojedacardenes@gmail.com; 6Department of Physiotherapy, University of Seville, C/Avicena s/n. 41009 Seville, Spain; emedrano@us.es

**Keywords:** manual lymphatic drainage, lymphatic vessels, breast cancer lymphedema, indocyanine green, lymphography

## Abstract

Manual Lymphatic Drainage (MLD) appears to stimulate lymphatic contraction, aid in the development of secondary derivation pathways, and stimulate the appearance of collateral pathways that could function as the main drainage routes of the limb in case of lymphedema. Through stretching, call up maneuvers are used to stimulate lymphangion reflex contraction and, therefore, lymphatic function. The aim was to describe the presence of areas and pathways of collateral lymphatic drainage under basal conditions and to determine, using Indocyanine Green (ICG) lymphography, whether an increase in these pathways occurs after 30 min of manual lymphatic stimulation with only call up maneuvers according to the Leduc Method^®®^. In this prospective analytical study (pretest–posttest), the frequency of presentation of areas and collateral lymphatic pathways was analyzed in 19 patients with secondary lymphedema of the upper limb after breast cancer using an infrared camera. Analyses were completed at three time points: after ICG injection, at baseline (pretest), and after the application of MLD (post-test). The Leduc Method maneuvers were applied to the supraclavicular and axillary nodes, chest, back, Mascagni, and Caplan pathways. The areas visualized in the pretest continued to be visible in the posttest. Additional pathways and fluorescent areas were observed after the maneuvers. The McNemar test showed statistical significance (*p* = 0.008), the odds ratio was infinite, and the Cohen’s g value was equal to 0.5. Manual stimulation by call up maneuvers increased the observation frequency of areas and collateral lymphatic pathways. Therefore, ICG lymphography appears to be a useful tool for bringing out the routes of collateral bypass in secondary lymphoedema after cancer treatment.

## 1. Introduction

Manual Lymphatic Drainage (MLD) increases the contractile activity of the lymphangion by gently pulling on its wall, sending the edematous fluid through the lymphatic dividing lines from the edematous area to neighboring lymphosomes, through the interstitial tissues areas, where the edema can be reabsorbed by healthy lymphatics [1,2,3]. It stimulates lymphatic contraction, develops the secondary derivation pathways [4], and stimulates the appearance of collateral pathways [5,6] that could function as the main drainage routes of the limb in cases of dysfunction [1,3].

The Leduc^®®^ drain massage uses two types of maneuvers: call ups or evacuation, and reabsorption or uptake. Through stretching, call up maneuvers are used to stimulate lymphangion reflex contraction and, therefore, the lymphatic collectors [7]. MLD sequences used to treat upper limb secondary lymphedema include stimulation of the neck regions; the healthy and affected axilla; the axillo-axillary pathway through the chest and back; the Mascagni and Caplan pathways; and depending on the method, a chest–groin path, as described by Kubik St [8].

In secondary lymphoedema, the Mascagni route is a lymphatic route of superficial derivation that may appear to extend from the upper extremity, joining the lymph node at the base of the neck, in two ways: directly to a lymph node of the neck (direct Mascagni), or indirectly (Mascagni Indirect) towards a clavicular lymph node and from this to a lymph node at the base of the neck [9]. The Caplan pathway may appear in a posterior stream from the arm; it follows the posterior deltotricipital sulcus and is directed to a ganglion of the posterior scapular chain [10].

Leduc et.al. [10] conducted a study on the drainage of the replacement routes in the upper extremity in over 300 human cadavers. By injecting a dye dissolved in turpentine, he detected numerous replacement pathways, mostly located in the posterior region of the shoulder, capable of evacuating lymph from the upper extremity. They stated that often, to allow the drainage replacement of the normal routes by collateral routes, it is necessary to facilitate the opening of these collaterals through increasing the hydrostatic pressure inside the lymphatic system by means of careful manipulations. This author, Caplan, I. and Lievens, P. [11], described the possible collateral substitution pathways after breast cancer treatment and found that they do not occur with the same frequency in all patients.

The normal structure and pathological function of the lymphatic system have been studied, using different imaging techniques, such as lymphoscintigraphy, magnetic resonance, ultrasound, Near-Infrared Fluorescence Imaging, Single-Photon Emission Computed Tomography/Computed Tomography, and Indocyanine Green (ICG) lymphography [12].

ICG lymphography, approved for use in humans in 1956 [13] and employed to obtain fluorescent imaging in subjects with secondary lymphoedema, from 2007 [14], involves injecting ICG, a highly fluorescent substance, into the subcutaneous space [15], which poses no risk of radioactive exposure [16]. As soon as it is injected, it binds with free local albumin [17] and local lipoproteins in the interstitial space. These new molecules are rapidly reabsorbed (≤30 min) [18] and selectively enter the lymphatic channels [17]. Using a specific camera with infrared diodes, the operator films the patient’s skin at between 15 and 30 cm [17]. It is possible to observe the superficial lymphatic architecture as a dynamic road map at a depth of 1 to 2 cm from the skin surface [17,18,19,20].

Thus, ICG lymphography or lympho-fluoroscopy with ICG is considered a useful, minimally invasive, and safe tool for obtaining images of the lymphatic system [21] and its collateral drainage pathways in vivo and in real time [19,22].

This technique has been used for the early detection and classification of edema [23]. Mihara et al. [24] believe that the use of fluorescent images is superior to lympho-scintigraphy, for lymphedema assessment. For this reason, ICG lymphography has been used to evaluate the contractile function of the lymphatic system before and after pneumatic compression therapy [21] or after the MLD technique [25].

Tashiro et al. [18] studied 192 lower limbs using ICG lymphography, and in three subjects, superficial lymphatic collateral vessels were found to be located above the umbilical level, reaching the ipsilateral axillary lymph nodes. Further, through this imaging technique, Giacalone [17] observed the ability of MLD to open the lymphatic pathways and visualized routes of Mascagni and Caplan. Similarly, Tashiro et al. [16], in another study, visualized accessory lymphatic pathways in the upper arm that extended to the shoulder, neck, and breast regions in patients with secondary lymphedema of the upper limb.

Our study aims to identify and describe the presence of collateral drainage areas and pathways, using ICG lymphography, in patients with secondary upper limb lymphedema after breast cancer treatment under baseline conditions (pretest) and to determine whether an increase in the presence of collateral drainage areas and pathways is observed after MLD treatment using the call up maneuvers described by the Leduc^®®^ method.

## 2. Methods

### 2.1. Study Cohort

This prospective analytical study (pretest–posttest) was carried out at the University Hospital of Gran Canaria Dr. Negrín (H.U.G.C. Dr. Negrín), Spain, between January and July 2017. The 19 upper limbs from 19 women diagnosed with secondary lymphedema of the upper limb after axillary lymph node resection for breast cancer were evaluated. The patients belonged to the waiting list for physical treatment of the Lymphatic Pathology Unit of the Rehabilitation Service of H.U.G.C. Dr. Negrín. The Committee of Ethics in Biomedical Research (CEIB) of the H.U.G.C. Dr. Negrín approved this study protocol (Code CEIC Negrín 170022), as well as the Spanish Agency of Medicines, which classified it as non-observational without medication.

All study participants freely accepted and signed the informed consent. Patients with clinical suspicion or presence of deep vein thrombosis, with allergy to iodine or some of its derivatives, and those who did not sign an informed consent were excluded.

A total of 94.7% of the patients received radiotherapy as a complementary treatment to axillary lymph node resection. The mean age of the patients was 59 years (53–68), with 8 of the 19 volunteers presenting with stage IIa and 11 with stage IIb lymphedema, according to the classification proposed by the International Society of Lymphology (ISL) [26]. The severity percentage calculated using the formula recommended by Jean Claude Ferrandez [27] was mild in 26% of subjects (*n* = 5), moderate in 37% (*n* = 7), and severe in 37% (*n* = 7).

### 2.2. Intervention

First, to avoid the pain and/or the itching that ICG may produce during injection, all the patients received a local anesthetic in the place of subsequent tracer injection [20]. After 5 minutes, 0.3 mL of 25 mg ICG (Verdye^®®^, Waas Anita, S.A. Diagnostic Green GmbH, Aschheim-Dornach, Germany) in 5 mL of glycosylated serum of 5% was injected into the second and fourth interdigital spaces in the hand of the affected member. After the injection, the patients were asked to remain still for 5 minutes and after that time to carry out isolated flexion and extension movements of the fingers of the affected limb for 5 minutes.

The subjects remained supine throughout the study, except during the observations of the posterior areas of the body, for which they were placed in lateral decubitus on the healthy side. The first observation, the baseline, was carried out by a specialized doctor and a physiotherapist, 45 min after ICG injection. ICG tracer presence was observed by an infrared camera (Photodynamic Eye, Hamamatsu Photonics K.K., Hamamatsu City, Shizuoka Pref., Japan) by tracing the uptake of the marker in the following regions: (1) Affected upper limb (shoulder: upper, anterior, posterior and lateral sides, and lymph vessels from the arm to the ipsilateral axilla). (2) Thorax (breast, intercostal space, and abdomen) in the healthy and affected side. (3) Back (supraclavicular and scapular spaces) in the healthy and affected side. (4) Ganglionar areas: axillary, parasternal, supraclavicular, scapular, and deltopectoral.

In the same way, the presence or absence of the main collateral lymphatic pathways was analyzed (affected axilla to ipsilateral scapula, affected back’s quadrant to ipsilateral supraespinal region, inter-axilla pathway, affected breast to abdominal region, affected breast to supraclavicular space, interscapular, and the Mascagni and Caplan´s pathways). All images were recorded for detailed evaluation.

After this initial evaluation, the physiotherapist carried out a 30-min manual lymphatic stimulation session in which only call up maneuvers were developed on the supraclavicular cervical and axillary bilateral lymph nodes. All possible collateral routes were also stimulated in the chest and back, in addition to the Mascagni and Caplan pathways [9].

Areas below the shoulder were not stimulated. After manual lymphatic stimulation, a second exploration was carried out, following the same methodology performed in the pretest. All the patients were contacted after 24 h to verify that no adverse effects had arisen. In order to avoid the contamination of the patient’s skin with the ICG (Figure 1) by the manipulation of the practitioner during tracer injection by the physiotherapist during drainage maneuvers, or by the patient himself, all individuals (participants, physicians, and physiotherapists) wore latex gloves.

### 2.3. Statistical Analysis

The McNemar test was used to determine if there were statistically significant differences between the observations at baseline (pretest) and those found after applying the MLD (post-test). The odds ratio and Cohen’s g value were calculated.

## 3. Results

### 3.1. Results Obtained at Baseline (Pretest)

Tracer presence was observed in the lymphatic pathways from the affected arm to the ipsilateral axilla in two cases. One of these patients also showed tracer presence in the affected breast; in the other, ICG could be visualized in the posterior region of the shoulder and the ipsilateral supraclavicular and scapular nodes (Table 1). No contralateral cases were observed.

### 3.2. Results Obtained after the Stimulation Session by Means of Call Up Maneuvers (Post-Test)

In addition, the following collateral pathways were identified (Table 1): A pathway from the affected breast to the ipsilateral abdominal region; a descending pathway through the abdomen from the chest in the healthy side; a new way from the affected arm to the affected axilla; seven areas in the shoulder: five of them in the posterior region; and three tracks from the anterior shoulder to the clavicular lymph node of the ipsilateral neck on the affected side (Mascagni) (Figure 2b);

The Figure 2a, made in the pretest, shows the absence of tracer in anterior region of the shoulder of one of the subjects. In Figure 2b the presence of the ICG tracer is visible, in the same case, after the manual lymphatic maneuvers following the Mascagni pathway.

In Figure 3, taken in the posttest, the migration of the tracer can be observed since the affected armpit to the ipsilateral scapular region.

In the Figure 4a. taken from the same subject, in the pretest, the absence of tracer ICG can be verified. A pathway from the affected scapula to the ipsilateral supraspinal region (Caplan) was visualized in the posttest (Figure 4b).

Finally, the McNemar statistical test showed a statistically significant difference between the findings obtained in the posttest compared to the pretest (*p* = 0.008). The Cohen’s g value was 0.5.

## 4. Discussion

An increase in ICG presence was observed in the posttest with respect to the pretest, after the application of call up maneuvers in the proximal regions of the shoulder, identifying not only physiological areas of drainage but also derivation or collateral pathways. At baseline, tracer uptake was always observed to follow derivation routes to areas of the ipsilateral region on the affected side (supraclavicular, infraclavicular, delto-pectoral, and scapular nodes), as described in the existing literature [16,17]. The observations recorded before the manual stimulation remained visible in the posttest or even showed an increased frequency of observation.

The McNemar test showed a statistically significant difference between the findings obtained in the post-test compared to the pretest (*p* = 0.008), and the Cohen’s g value was 0.5. Thus, the size of the observed effect was large [28].

After manual stimulation, the case observed in the posterior shoulder was still visible, and five new cases were recorded. These findings support the observations of Leduc [10] who also suggests the posterior region of the shoulder to be preferential for upper limb drainage. Our findings contrast with those of Tashiro [25], who confirmed a greater presence of the tracer in the anterior shoulder region. This difference in observed results could be due to the manual stimulation carried out in this region in our study.

According to Leduc [10], the increase in the presence of ICG observed in the posttest compared to the pretest suggests that MLD manipulations may induce the opening of derivation pathways. As observed by Suami [29] in canine models, the opening of collaterals could occur as a compensatory mechanism after surgery. In our study, after stimulation by call up maneuvers in the proximal regions, we verified the presence of new collateral pathways. ICG presence in three Mascagni pathways, four armpits, and one Caplan pathway was confirmed. This is similar to the observations of Giacalone [17], who identified s fluorescent tracer in one armpit and a Mascagni and a Caplan route after a session of MLD. The greater frequency of presentation observed in our study could be attributed to the methodological differences in manual stimulation interventions. While Giacalone [17] carried out a 20-minute drainage session, following the Leduc^®®^ Method, addressing the proximal regions and the edematous area with call up and resorption maneuvers, our intervention consisted of 30-min manual stimulation of only the proximal areas using the call up maneuvers. This increase in proximal area stimulations could have favored the visualization of a greater number of cases.

Our study findings confirm that the appearance of substitution pathways or collateral drainage pathways do not occur with the same frequency at baseline [10] nor after MLD [7,9], nor in the same way, in all patients with secondary upper limb lymphedema post breast cancer treatment. The individual characteristics and the state of the lymphatic system after breast cancer treatment could influence the presentation of the derivation routes.

We would also like to highlight the two routes observed, one located under the affected chest and the other under the healthy chest that descended towards the abdomen, as these were the only contralateral collateral pathways observed in our study. These descending pathways, located on each side of the thorax, would connect the upper regions of the trunk with the inguinal nodes as described by Kubik [8]. We have not found reference to this finding in other studies [7,30]. Tashiro [18] identified some pathways in the thorax, but these were directed from the abdominal region upwards towards the armpits in lower limb lymphedema.

In view of our results, we agree with other authors [15,16,21] that ICG lymphography is useful as a simple and economical technique for common clinical study of the lymphatic system. In accordance with Giacalone [17], we consider it essential that physiotherapists and other professionals involved in physical or surgical treatment know the “personal” map of the derivation routes and the possibilities of collateral evacuation for each patient. Our results suggest that manual stimulation of the proximal regions and collateral pathways by means of call up maneuvers could facilitate the observation of these drainage routes and the mapping of the functional pathways of each patient, allowing the development of personalized treatment guidelines to the characteristics observed in all patients. The individualized drainage sequences would increase the effectiveness of the maneuvers and optimize treatment times.

It would have been interesting to include a control group in this study, but to recruit it at the time it was done was impossible due to technical difficulties. On the other hand, we highlight the need for studies with a larger sample size to ascertain the form and frequency of the presentation of evacuation routes at baseline as well as the analysis of certain factors that could influence the presence or absence of these collateral pathways, such as the realization of previous physiotherapy treatments or radiotherapy.

Similarly, it would be interesting to determine whether manual lymphatic drainage can open these bypass pathways permanently or if, on the contrary, they are active only while they are stimulated, in which case, spontaneous edema improvement could not be expected and would justify the need to repeat MLD treatments in case of edema destabilization.

## 5. Conclusions

In our study, under basal conditions, it was possible to visualize ICG markers only in two patients in the posterior shoulder, supraclavicular and scapular ganglia, affected breast, and two pathways that were directed from the arm to the affected axilla. The findings in the pretest were always located in ipsilateral regions of the affected side.

After 30 min of manual stimulation physiotherapy intervention of the proximal regions using the Leduc^®®^ method call up maneuvers, the fluorescent areas recorded in the pretest remained visible but with increased frequency of visualization. In addition, new areas and collateral drainage pathways were observed. Regarding the marking of contralateral regions to the affected side, only in the post-test was a descending path from the lower part of the healthy chest to the ipsilateral abdomen identified.

MLD manipulations of only the proximal areas using the call up maneuvers induce the opening of derivation pathways but not with the same frequency or in the same way in all patients. For the success of any therapeutic strategy, it is essential to know the “personal” map of the derivation routes and the possibilities of collateral evacuation for each patient

## Figures and Tables

**Figure 1 jcm-08-01917-f001:**
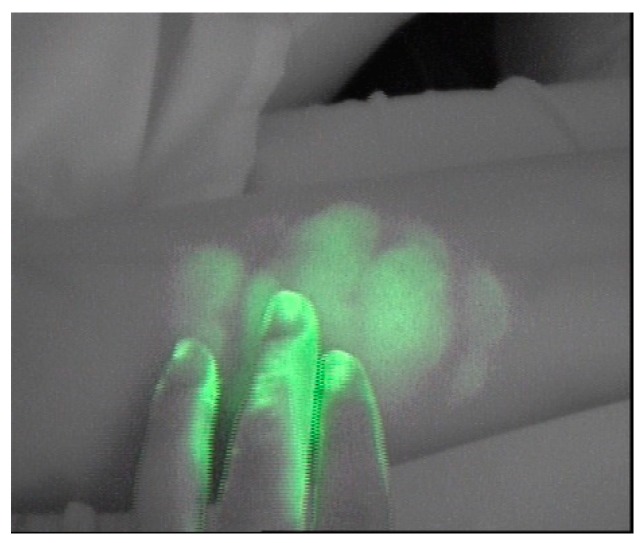
Contamination of the patient and physiotherapist skin with ICG.

**Figure 2 jcm-08-01917-f002:**
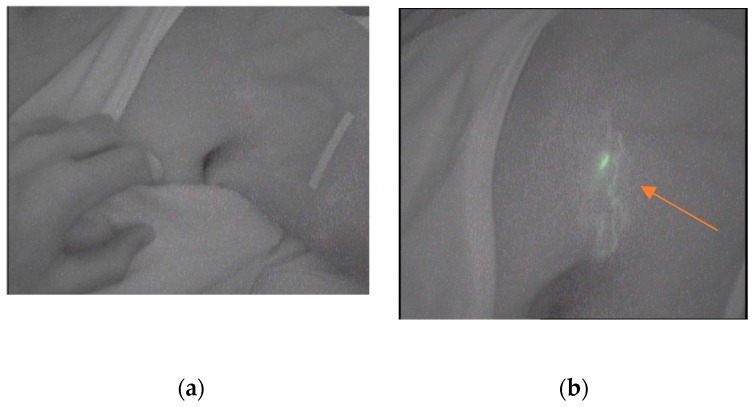
(**a**) Anterior vision of the affected shoulder (pretest); (**b**) Pathway from the shoulder to the clavicular lymph node (posttest).

**Figure 3 jcm-08-01917-f003:**
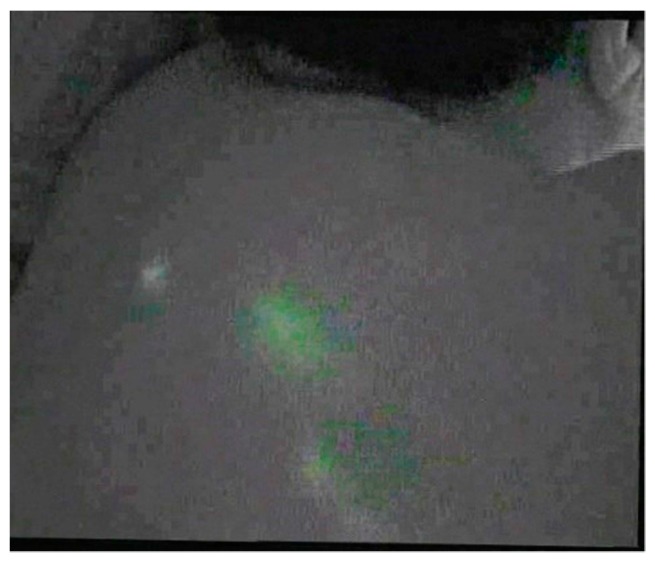
A pathway from the affected axilla to the ipsilateral scapular region.

**Figure 4 jcm-08-01917-f004:**
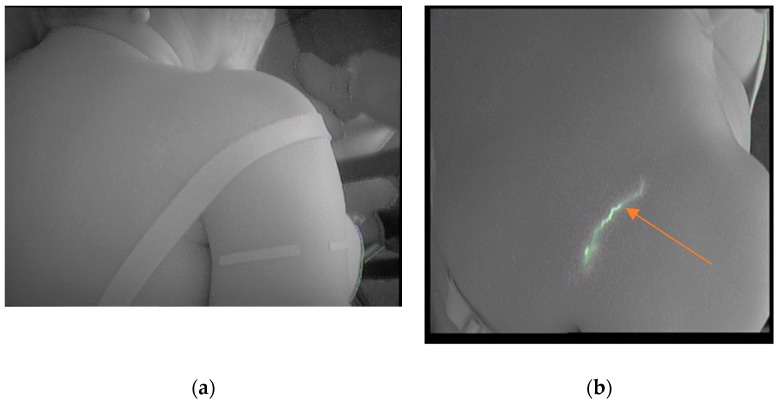
(**a**): Visualization of the scapular area pretest; (**b**) Pathway from the affected scapula to ipsilateral supraespinal region (postest).

**Table 1 jcm-08-01917-t001:** Observations of the uptake of tracer in the pretest and posttest.

	PRE-TEST	POST-TEST
**Shoulder region:**		
Anterior	0	1
Posterior	1	6
Superior	0	1
**Lymph nodes´ areas:**		
Ipsilateral axilla	0	4
Supraclavicular	1	4
Ipsilateral infraclavicular	0	1
Delto-pectoral	0	2
Scapulars	1	5
**Other areas:**		
Affected breast	1	2
Arm way affected to affected axilla	2	3

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
