# Peer review of "Visualization of Accessory Lymphatic Pathways, before and after Manual Drainage, in Secondary Upper Limb Lymphedema Using Indocyanine Green Lymphography"

_jcm, 2019, doi:10.3390/jcm8111917_

Round 1
Reviewer 1 Report
The authors map ICG clearance from the upper limbs of 19 patients with breast cancer related lymphedema before and after a 30-minute manual lymphatic drainage session focused on proximal “call up” maneuvers. Not surprisingly, the authors show increased drainage to the axilla/shoulder after the MLD. However, no control group that did not received MLD was used, so no real conclusions can be drawn. With more time after ICG injection, it is expected to see ICG further from the injection site. So it is not clear if the results are due to the time that passed between the imaging sessions, due to MLD or due to some combination. Further, the quantification and statistical tests do not appear rigorous. Further, the manuscript is very difficult to read as a result of poor English. Although, I understand the importance of tailoring MLD for individual patients and that ICG imaging may provide a tool in order to personalize MLD, the quality of this manuscript needs to improve in order to advance this paradigm.
Comments:
Abstract: “To describe the presence of areas and pathways of collateral lymphatic drainage under basal conditions and to determine, using Indocyanine Green (ICG) lymphography, whether an increase in these pathways occurs, after 30 min of manual lymphatic stimulation with only call up maneuvers according to the Leduc Method®®.” This is not a sentence. Please correct for grammar. There are many other instances in the abstract and text where grammar and punctuation needs to be corrected and improved. The paper needs serious editing for clarity and common English usage.
Abstract: “The areas visualized in the pretest continued to be visible while additional pathways and fluorescent areas were also observed.” Are the additional pathways observed after the maneuvers? This sentence is not clear.
Lines 43-45: “Manual Lymphatic Drainage (MLD) increases the contractile activity of the lymphangion by gently pulling on its wall, sending the edematous fluid through the lymphatic dividing lines from the edematous area to neighboring areas where the edema can be reabsorbed by healthy lymphatics” This does not make logical sense. I agree that MLD may increase the contractile activity and it might be able to move fluid into a lymphosome (lymphatic drainage area) with healthy lymphatic vessels for removal. However, MLD does not move fluid from the inactive lymphosome to an active one through collecting lymphatic vessels, but through the interstitial tissues. Therefore, this sentence will mislead the readers.
Paragraph beginning on line 55: It seems inappropriate to use the word ganglia to describe the route of lymph flow. Ganglia are structures that refer to nerves, so using it here is confusing, or if taken literally, just plain incorrect.
No control group that did not received MLD was used, so no real conclusions can be drawn. With more time after ICG injection, it is expected to see ICG further from the injection site. So it is not clear if the results presented are due to the time that passed between the imaging sessions, due to MLD or due to some combination.
Lines 192-194: “Finally, the McNemar statistical test showed a statistically significant difference between the findings obtained in the post-test compared to the pretest (p = 0.008). The odds ratio was infinite, and the Cohen’s g value was 0.5.” It is not at all clear what was tested statistically. Did the authors combine all of the anatomical locations as a single parameter? This does not seem appropriate as this will over represent individual patients. Further as the patient number is only 19, there is no way for the b+c to be greater than 25, so some correction is needed to model a binomial distribution instead of a chi-squared distribution. Care must also be taken to properly pair each patient. It is also not clear how the odds ratio can be infinite.
2 out of 19 patients had ICG drainage from the arm to the axilla/shoulder pre-MLD. How many patients had ICG to from the arm to the axilla/shoulder post-MLD?
The term Ipsilateral is more standard than the term homolateral. Please change to ipsilateral throughout the manuscript.
Author Response
Thanks to you and the editors for their careful assessment of our manuscript. We´re appreciated your ideas and are hoping that we were able to meet all your concerns adequately. Our response its italicized, manuscript changes are highlighted in gray.
First review report.
Dear Review, we apologize for our English. It has been checked for correct use of grammar and common technical terms, and edited to a level suitable for reporting research in a scholarly journal by native English speaking editors from MDPI editing service. You can check the changes below;
Abstract
The aim was . Added (page 1 , line 24).
“On” to “in” . Corrected ( page 1, line 29).
Applied. Added (page1, line 32).
Introduction
“Other” to “another”. Corrected (page3, line 94).
Methods
The 19 upper limbs from 19 women diagnosed with secondary lymphedema of the upper limb after axillary lymph node resection for breast cancer were evaluated. Reworded (page 3, lines 106-108).
Results
Were identified. Corrected (page 6, line 179).
“Ganglia” to” lymph node”. Corrected (page 6 , line 188).
“Ganglionic areas” to “lymph nodes ‘areas” ( page 5, Table1) .
Discussion
“20” to “twenty”. Corrected (page 7, line 232).
Conclusions
“Nor” to “not” . Corrected (page 9, line 283).
For the success of any therapeutic strategy, it is essential to know the “personal” map of the derivation routes and the possibilities of collateral evacuation for each patient. Reworded (page 9, lines 283-285).
Reviewer comments:
Abstract:
“To describe the presence of areas and pathways of collateral lymphatic drainage under basal conditions and to determine, using Indocyanine Green (ICG) lymphography, whether an increase in these pathways occurs, after 30 min of manual lymphatic stimulation with only call up maneuvers according to the Leduc Method®®.” This is not a sentence.
We added in the manuscript: “the aim was” ( page 1 , line 24)
“Please correct for grammar. There are many other instances in the abstract and text where grammar and punctuation needs to be corrected and improved. The paper needs serious editing for clarity and common English usage.”
English has been checked for correct use of grammar and common technical terms, and edited to a level suitable for reporting research in a scholarly journal by native English speaking editors from MDPI editing service.
“The areas visualized in the pretest continued to be visible while additional pathways and fluorescent areas were also observed.” Are the additional pathways observed after the maneuvers? This sentence is not clear.
We reworded this sentence (page 1, line 32-34). The areas visualized in the pretest continued to be visible in the postest. Additional pathways and fluorescent areas were observed after the maneuvers.
Introduction:
“Lines 43-45: “Manual Lymphatic Drainage (MLD) increases the contractile activity of the lymphangion by gently pulling on its wall, sending the edematous fluid through the lymphatic dividing lines from the edematous area to neighboring areas where the edema can be reabsorbed by healthy lymphatics” This does not make logical sense. I agree that MLD may increase the contractile activity and it might be able to move fluid into a lymphosome (lymphatic drainage area) with healthy lymphatic vessels for removal. However, MLD does not move fluid from the inactive lymphosome to an active one through collecting lymphatic vessels, but through the interstitial tissues. Therefore, this sentence will mislead the readers.”
In order to clarify this concept, we added to the sense: neighboring lymphosomes, through the interstitial tissues ( page2, line 45)
“Paragraph beginning on line 55: It seems inappropriate to use the word ganglia to describe the route of lymph flow. Ganglia are structures that refer to nerves, so using it here is confusing, or if taken literally, just plain incorrect.”
We corrected “ganglia” to “ lymph node”. (Page 2, lines 57-58)
We reworded this paragraph to make it more understandable. The Mascagni´s pathways were described as follows: directly to a lymph node of the neck (direct Mascagni), or indirectly (Mascagni Indirect) towards a clavicular lymph node and from this to a lymph node at the base of the neck [9]. (Page 2, lines 57-58)
Methods
“No control group that did not received MLD was used, so no real conclusions can be drawn. With more time after ICG injection, it is expected to see ICG further from the injection site. So, it is not clear if the results presented are due to the time that passed between the imaging sessions, due to MLD or due to some combination.”
Dear editor, we sincerely appreciate your wise appreciations and careful comments about our study. Indeed, we fully agree with your approach, to include a control group would have been interesting, but to recruit it at the time the study was done, was impossible due to technical and logistical difficulties.
We highlighted this issue in the Discussion of our manuscript.( page 8, lines 259-261)
Results
“Lines 192-194: “Finally, the McNemar statistical test showed a statistically significant difference between the findings obtained in the post-test compared to the pretest (p = 0.008). The odds ratio was infinite, and the Cohen’s g value was 0.5.”
“It is not at all clear what was tested statistically. Did the authors combine all of the anatomical locations as a single parameter? This does not seem appropriate as this will over represent individual patients”.
The registration of the observed areas described in the methodology, in a more extensive way after the recommendations received, was carried out systematically in each and every one of the study participants. Added (page 4 lines 136-148)
As shown in the table below the number of subjects who presented spontaneously, in the pretest, ICG tracer in the anatomical regions studied was recorded. They were a total of 3 cases.
|
No observation post |
Yes observation post |
Total |
No observation pre |
7 (43,75%) a |
9 (56,25%) b |
16 |
Yes observation pre |
0 (0.0%) c |
3 (100.0%) d |
3 |
Total |
7 |
12 |
19 |
After the intervention was performed (manual drainage of the proximal anatomical regions), not only were the routes observed in the three subjects of the pretest still visible, but also 9 new subjects presented ICG tracer in some of the anatomical areas under observation.
Further as the patient number is only 19, there is no way for the b+c to be greater than 25, so some correction is needed to model a binomial distribution instead of a chi-squared distribution.
To compare if there were significant differences between the number of patients who were seen pathways before and after exposure ( man), the McNemar test (with continuity correction) was used for paired data, where a p-value = 0.008 was obtained.
We attach the screenshot, the McNemar’s test with continuity correction, p-val = 0.007661, rounded: p-val = 0.008.
“Care must also be taken to properly pair each patient. It is also not clear how the odds ratio can be infinite.”
Thank you very much for this important explanation. Next, we will try to briefly explain the calculations made on which we base our assertion.
After to McNemar test, the effect size was calculated, we use the “rcompanion” package of RStudio program, function “cohenG” (Cohen’s g and odds ratio for paired contingency tables, page 17/99 from the package “rcompanion” pdf) in order to calculate OR and effect size
This is the link from the “rcompanion” package of RStudio.
We attach the screenshot, where the results are: OR = infinitve and Cohen’s G = 0.5:
Besides, the information that we can extract from the pdf, in the following link also appears who to calculate it:
https://rcompanion.org/handbook/H_05.html
Considering a 2 x 2 table, with (a) and (d) being the concordant cells and (b) and (c) being the discordant cells, the OR is simply the greater of (b/c) or (c/b):
OR = max (9/0, 0/9) = 9/0 = infinite
In any case, and in order to avoid confusion, we have deleted the sentence “The odds ratio was infinite”.
“2 out of 19 patients had ICG drainage from the arm to the axilla/shoulder pre-MLD. How many patients had ICG to from the arm to the axilla/shoulder post-MLD?”
We are sorry for missing this relevant information.
A new way from the affected arm to the affected axilla; Seven areas were market in the shoulder: 5 in the posterior region.We added it in page 6, lines 184-187
“The term Ipsilateral is more standard than the term homolateral. Please change to ipsilateral throughout the manuscript.”
We corrected “homolateral” to ipsilateral and proofread the manuscript thoroughly.

Reviewer 2 Report
Interesting study where the authors observed and reported about collateral lymphatic pathways with ICG. It is well known that superficial lymphatic anatomy varies within each individual, but this is a nice demonstration of where these differences might influence patient outcome. I'm not sure if Table 1 describes the number of vessels counted? Images pre and post of the same patient would be worthwhile - perhaps with some arrows and clear description?
We know very little about healthy controls and their lymphatic architecture. For future experiments comparing with a control group would be interesting.
Author Response
Thanks to you and the editors for their careful assessment of our manuscript. We´re appreciated your ideas and are hoping that we were able to meet all your concerns adequately. Our response its italicized, manuscript changes are highlighted in gray.
General comments.
“Interesting study where the authors observed and reported about collateral lymphatic pathways with ICG. It is well known that superficial lymphatic anatomy varies within each individual, but this is a nice demonstration of where these differences might influence patient outcome.”
Dear reviewer, first of all, thank you for your comments and valuable suggestions.
“I'm not sure if Table 1 describes the number of vessels counted.”
In fact, Table 1 shows only the pre and postest observation (areas, vessels, pathways, lymph nodes), but not all the regions studied. To clarify this issue, we completed a detailed description of all the regions analyzed in methodology of the manuscript (Page 4, lines 136- 147).
“Images pre and post of the same patient would be worthwhile - perhaps with some arrows and clear description?”
The title of Fig. 1B (before Fig. 1) and Fig. 4B (before Fig. 4) were modified and two images were added (Figures 2A and 4A), in order to show the pretest-posttest situation. The relevant changes in posttest figures have been indicated with arrows. (Page 5)
We apologize for not being able to include the pretest image in Figure 3. Pathway from the affected axilla to ipsilateral scapular region; the resolution of it is very poor.
“We know very little about healthy controls and their lymphatic architecture. For future experiments comparing with a control group would be interesting.”
We agree with your comment; the lymphatic system is a great unknown. In a normal situation, without lymph node resection, it could present some specific difference between subjects, but the truth is that the study of their behaviour in a pathological situation is very complex. In principle we could expect it to behave similar or to circulate in a standardized way, but the findings of our study demonstrate once again the variability between individuals in the pretest study. To include a control group in this study would have been interesting, but it was impossible to recruit a control group due to technical and logistical reasons.
We highlighted this issue in the Discussion of our manuscript .( page 8, lines 259-261)